# High seroprevalence of Immunoglobulin G (IgG) and IgM antibodies to SARS-CoV-2 in asymptomatic and symptomatic individuals amidst vaccination roll-out in western Kenya

Shehu Shagari Awandu[1,2]*, Alfred Ochieng Ochieng[2], Benson Onyango[2], Richard Odongo Magwanga[2,3], Pamela Were[2], Angeline Atieno Ochung'[2], Fredrick Okumu[2], Marceline Adhiambo Oloo[2], Jim Seth Katieno[4], Shirley Lidechi[4], Fredrick Ogutu[5], Dorothy Awuor[2], Joy Nyangasi Kirungu[3], Francis Orata[6], Justine Achieng[5], Bonface Oure[5], Regina Nyunja[2], Eric M. O. Muok[4], Stephen Munga[4], Benson Estambale[1]*

1 School of Health Sciences, Jaramogi Oginga Odinga University of Science and Technology, Bondo, Kenya, 2 School of Biological, Physical, Mathematics and Actuarial Sciences, Jaramogi Oginga Odinga University of Science and Technology, Bondo, Kenya, 3 State Key Laboratory of Cotton Biology/Institute of Cotton Research, Chinese Academy of Agricultural Science, Anyang, Henan, China, 4 Kenya Medical Research Institute Centre for Global Health Research (CGHR), Kisumu, Kenya, 5 Kenya Industrial Research and Development Institute (KIRDI), Kisumu, Kenya, 6 Masinde Muliro University of Science and Technology (MMUST), Kakamega, Kenya

* sawandu@jo_oust.ac.ke (SSA); bestambale@jooust.ac.ke (BE)

## Abstract

The population's antibody response is a key factor in comprehending SARS-CoV-2 epidemiology. This is especially important in African settings where COVID-19 impact, and vaccination rates are relatively low. This study aimed at characterizing the Immunoglobulin G (IgG) and Immunoglobulin M (IgM) in both SARS-CoV-2 asymptomatic and symptomatic individuals in Kisumu and Siaya counties in western Kenya using enzyme linked immunosorbent assays. The IgG and IgM overall seroprevalence in 98 symptomatic and asymptomatic individuals in western Kenya between December 2021-March 2022 was 76.5% (95% CI = 66.9–84.5) and 29.6% (95% CI = 20.8–39.7) respectively. In terms of gender, males had slightly higher IgG positivity 87.5% (35/40) than females 68.9% (40/58). Amidst the ongoing vaccination roll-out during the study period, over half of the study participants (55.1%, 95% CI = 44.7–65.2) had not received any vaccine. About one third, (31.6%, 95% CI = 22.6–41.8) of the study participants had been fully vaccinated, with close to a quarter (13.3% 95% CI = 7.26–21.6) partially vaccinated. When considering the vaccination status and seroprevalence, out of the 31 fully vaccinated individuals, IgG seropositivity was 81.1% (95% CI = 70.2–96.3) and IgM seropositivity was 35.5% (95% CI = 19.22–54.6). Out of the participants that had not been vaccinated at all, IgG seroprevalence was 70.4% (95% CI 56.4–82.0) with 20.4% (95% CI 10.6–33.5) seropositivity for IgM antibodies. On PCR testing, 33.7% were positive, with 66.3% negative. The 32 positive individuals included 12(37.5%) fully vaccinated, 8(25%) partially vaccinated and 12(37.5%) unvaccinated. SARs-CoV-2 PCR positivity did not significantly predict IgG (p = 0.469 [95% CI 0.514–4.230]) and IgM (p = 0.964

**Data Availability Statement:** All relevant data are within the paper and its Supporting Information files.

**Funding:** Funding was provided by the National Research Foundation-South Africa under the COVID-19 Africa Rapid Grant Fund (Nr: COV19200616532700). SSA is supported by AREF Research Development Fellowship 2022, (AREF-312-AWAN-F-C0907). The funders had no role in study design, data collection and analysis, decision to publish, preparation of the manuscript.

**Competing interests:** The authors have declared that no competing interests exist.

[95% CI 0.380–2.516]) positivity. These data indicate a high seroprevalence of antibodies to SARS-CoV-2 in western Kenya. This suggests that a larger fraction of the population was infected with SARS-CoV-2 within the defined period than what PCR testing could cover.

## Introduction

The coronavirus disease 2019 (COVID-19) caused by severe acute respiratory syndrome coronavirus 2 (SARS-CoV-2), has infected more than half a billion people globally [1]. The COVID-19 pandemic continues to disrupt lives, increase mortality in people with underlying co-morbidities and severely impact world economies [2]. While all continents have been severely impacted, Africa has registered low scores on major metrics including mortality rates, number of cases and absence of exponential growth as predicted [3]. However, the reasons for this are still unclear. Exposure of the population to many infectious diseases in the continent, generating cross reactive protective antibodies is suggested as contributing to reduced severity to the infection [3–5].

The SARS-CoV-2 infection is associated with the development of a robust humoral immune response with variable levels of Immunoglobulin A (IgA), IgM and IgG isotypes as the infection progresses [6, 7]. Upon SARS-CoV-2 infection, the IgM response is quick and short-lived, detectable up to 20 days post infection and then wanes [7, 8]. In contrast, the IgG antibody responses peak after 25 days, and are more long lived and detectable up to 120 days post symptom onset [7, 8].

The kinetics of anti-SARS-CoV-2 especially IgG and IgM antibodies have been profiled in several epidemiological settings in Kenya [9]. Whilst an earlier study among blood donors found an overall IgG seroprevalence of 4.3% peaking in 35–44 year olds [10], in contrast a study among community health workers reported 20.8% seroprevalence [11]. More recently, a population survey in Nairobi recorded a 34.7% seroprevalence [12]. Majority of studies in Kenya thus far have mainly focused on the most at-risk population in both urban and rural areas of the country. The differences in seroprevalence makes it unclear whether the antibody response to SARS-CoV-2 in western Kenya, ravaged by a host of infectious diseases including malaria, HIV and tuberculosis is similar to the rest of the country [13].

Here, we examined the levels of IgM and IgG antibodies to SARS-CoV-2 in asymptomatic and symptomatic individuals amidst vaccination roll-out, in Kisumu and Siaya counties in western Kenya. We hypothesized that in western Kenyan populations burdened by several other infectious diseases, the COVID-19 antibody responses are not different in vaccinated and non-vaccinated individuals.

## Materials and methods

### Study design and participants

We screened and recruited individuals presenting to Kisumu and Siaya Counties referral hospitals for routine COVID-19 tests in western Kenya. All patients, regardless of COVID-19 symptoms were eligible for enrollment. Study procedures were explained to them, and an informed consent form signed by the participants. A detailed personal history and physical examination were carried out by the study doctor and documented on a predesigned form. Demographic data including age, gender, county of residence, symptoms, date of onset, severity, vaccination status and test type (PCR or antigen test), whether initial or follow-up/repeat.

## Sample size calculations

Sample size was calculated in an online platform http://www.raosoft.com/samplesize.html, using a margin of error of 9.78% and with a 95% confidence interval with a 50% response distribution, giving at least 96 samples.

## Sample collections

Participants provided stool and nasopharyngeal samples in viral transport media (AB Medical Inc). Additionally, participants provided a 5 ml venous blood sample, in sterile EDTA tubes, that was centrifuged to separate plasma and buffy coat. All the samples were transported under cold chain to Kenya Medical Research Institute, Centre for Global Health Research (CGHR).

## Laboratory assays

**Enzyme linked immunosorbent assay (ELISA).**   To detect the presence of IgG and IgM antibodies against SARS-CoV-2 S proteins respectively, serological assays were performed using the qualitative indirect SCoV-2 Detect™ IgG ELISA kit and SCoV-2 Detect™ IgM ELISA kit (InBios International, Seattle, USA). Briefly, 50 μL each of serum samples, positive, negative and cut-off controls in duplicates were added into the SCoV-2 Antigen coated microtiter ELISA plates. The plates were covered with parafilm and incubated at 37˚C for 1 hour in an incubator. The plates were subsequently washed 6 times using 300 μL of 1X Wash Buffer. 50 μL of conjugate was then added to the wells, plate covered with parafilm and incubated at 37˚C for 30 minutes in an incubator. The plates were washed 6 times using 300 μL of 1X wash buffer. 75 μL of Liquid TMB substrate was added into all wells and the uncovered plates incubated at room temperature in the dark for 20 minutes. Finally, 50 μL of stop solution was added per well and the plates incubated at room temperature for 1 minute. The plates were read on a BIOTEK ELX 800 absorbance microplate reader at 450 nm optical density. The raw optical densities (ODs) were recorded, and ratios computed by dividing the sample OD by the median OD of the assay cut-off control. The kit cut-off control provided by the manufacturer aids in monitoring the integrity of the kit and estimating the proper threshold to determine sample status. The cut-off is set up in triplicate with each assay. The control OD has to be greater than the negative control OD for an assay to be validated. Samples with IgG or IgM ratio greater than or equal to 1.1 considered positive and IgG or IgM ratio less than or equal to 0.9 considered negative.

## RNA extraction and COVID-19 PCR tests

Total nucleic acid from Nasopharyngeal samples in Viral Transport Medium (VTM) were extracted using QIAamp Viral RNA Kit (Qiagen) following manufacturer's instructions. The extracted RNA from the samples was stored at −20˚C awaiting SARS-CoV-2 RT-PCR. Real Time PCR was conducted using a DaAn Gene SARS-CoV-2 PCR kit (DaAn Gene Co, Ltd., of Sun Yat-sen University, China) as per manufacturer's instructions. The master mix was prepared by mixing 17 μl of NC (ORF1ab/N) PCR liquid A (reaction mix) and 3 μl of NC (ORF1ab/N) PCR reaction liquid B (enzyme), then 5 μl of the extracted sample was added to make the PCRs final volume of 25 μl in a PCR plate on a cold block. The PCR tubes were immediately transferred to an ABI 7500 Fast RT-PCR machine (Applied Biosystems) for detection of SARS-CoV-2. The probe detection modes were set as: ORF1ab: VIC, Quencher: NONE, N-Gene: FAM, Quencher: NONE, Internal Control: Cy5, Quencher: NONE, Passive reference: NONE. The PCR cycle was carried out on the following conditions: 1 cycle of 15 min at 50˚C, 1 cycle of 15 min at 95˚C, and 45 cycles of 94˚C for 15 s and 55˚C for 45 s. Results

were analyzed by 7500 Fast Real Time PCR software version 2.3 to identify SARS-CoV-2 positive targets by evaluating PCR curves for sigmoidal amplification. A sample was considered positive for the targeted pathogen when it had cycle threshold (CT) value within 38 cycles (Ct < 38) for both ORF1ab and N-Gene. For quality control, a positive and negative control was included in each run and validated prior to specific patient sample analysis. Additionally, a negative extraction blank which was a known negative specimen was included in each run to check for cross contamination and specificity.

## Statistical analyses

Seroprevalence was determined as positivity for either IgG or IgM subtypes over the total number of individuals tested. The association between SARs-CoV-2 qPCR results, gender, IgM and IgG was tested using binomial logistic regression analysis. All the statistical analyses were conducted in STATA Version 16.

## Ethical considerations

Ethical approval for the study was granted by Jaramogi Oginga Odinga University of Science and Technology (ERC/21/5/21-4), and a research license granted from Kenya National Commission of Science and Technology (NACOSTI/P/22/17545). Administrative approval was provided by the county governments of Kisumu and Siaya. All participants provided written informed consent or assent before enrollment.

# Results

## Demographics

A total of 98 participants were recruited into the study, with slightly more females 58.2% (58/98) than males 40.8% (40/98). The median age was 29 years (interquartile range 19–44) years. About 26.5% (26/98) of participants were asymptomatic, however, they were referred to the hospital for surveillance testing as some were contacts of people infected with SARS-CoV-2. Majority of the participants were symptomatic 73.5% (72/98), reporting multiple symptoms including history of fever, muscular pain, shortness of breath, headache, and sore throat amongst others. On severity of symptoms, 6.1% (6/98) reported mild symptoms with more than half 57.4% (56/98) reporting severe symptoms. The samples were distributed equally with 49 from Kisumu County and another 49 from Siaya County referral hospital. Before the study period from December 2019 to November 2021 from the general population, Kisumu County PCR tested 26,166 individuals, with 22,127 negative, 4,011 positive and 28 indeterminate. In contrast, Siaya County PCR tested 939 individuals, with 665 negative and 274 positive. During the study period from December 2021 to March 2022 from the general population, Kisumu County PCR tested 11,250 individuals with 10,621 negative, 627 positive and 2 indeterminate. In contrast, Siaya county tested 705 individuals, with 656 negative, 34 positive and 15 indeterminate [14].

## Seroprevalence

During the 3 months' duration from December 2021 to February 2022, the IgG and IgM overall seroprevalence in 98 symptomatic and asymptomatic individuals in western Kenya was 76.5% (95% CI = 66.9–84.5) and 29.6% (95% CI = 20.8–39.7) respectively. In terms of gender, males had slightly higher IgG positivity 87.5% (35/40) than females 68.9% (40/58) (Table 1). We compared the levels of SARS-CoV-2 IgG and IgM antibodies in Kisumu which is largely urban town and Siaya a more rural set up (Table 1). Whilst the IgG antibodies levels, were

**Table 1. Demographic characteristics, antibody responses and SARS-CoV-2 PCR results among study participants.**

| Characteristics | Total samples tested | IgG Seroprevalence (95% CI) | IgM Seroprevalence (95% CI) | PCR positivity* (%) |
|---|---|---|---|---|
| **Overall** | 98 | 76.5% (95% CI = 66.9–84.5) | 29.6% (95% CI = 20.8–39.7) | 32(33.7) |
| Kisumu County | 49 | 73.5(58.9–85.0) | 20.4(10.2–34.3) | 12(24.5) |
| Siaya County | 49 | 79.6(65.7–89.8) | 38.8(25.2–53.8) | 20(43.8) |
| **Sex** | | | | |
| Female | 58 | 68.9% (55.4–80.4) | 22.4(12.5–35.8) | 21(36.8) |
| Male | 40 | 87.5% (73.2–95.8) | 40.0(24.9–56.7) | 11(28.9) |
| **Age group in years** | | | | |
| 0–11 | 6 | 66.7(22.3–95.7) | 33.4(43.2–77.7) | 1(16.7) |
| 12–17 | 18 | 83.3(58.5–96.4) | 27.8(9.7–53.5) | 2(11.11) |
| 18–49 | 51 | 66.7(52.1–79.2) | 25.5(14.3–39.6) | 19(38.0) |
| 50–64 | 14 | 92.9(66.1–99.8) | 42.9(17.7–71.1) | 8(57.14) |
| >65 | 9 | 100(66.4–100) | 33.3(7.5–70.1) | 2(28.6) |

*PCR results were only available for 95 participants while serology outcomes were available for 98 participants.

almost similar in the two counties, IgM antibodies were more pronounced in Siaya (38.8%) than Kisumu (20.4%) respectively.

## Seroprevalence by age

To assess immune response to SARS-CoV-2 among asymptomatic and symptomatic individuals, we tested for their IgG and IgM antibody levels. We further stratified the individuals into several age groups and compared the responses based on gender. Participants aged 18–49 years had the highest levels of detectable IgG antibodies from either gender. While all adult males aged 50–64 years and those over 65 years were all IgG seropositive, all young females aged between 0–11 and adults over 65 years were IgG seropositive (Fig 1).

The detectable IgM antibodies were highest in participants aged 18–49 years and lowest in children aged between 0–11 and adults over 65 years though at lower frequency than IgG. Interestingly, all female participants aged 65 years and above were negative for IgM antibodies (Fig 2).

## Seroprevalence and vaccination status

Amidst the ongoing vaccination roll-out during the study period almost one third, (31.6%, 95% CI = 22.6–41.8) of the study participants had been fully vaccinated, with close to a quarter (13.3%, 95% CI = 7.26–21.6) partially vaccinated. In contrast over half of the study participants (55.1%, 95% CI = 44.7–65.2) had not received any vaccine (Fig 3). When considering vaccination status and seroprevalence, out of the 31 fully vaccinated individuals, IgG seropositivity was 87.1% (95% CI = 70.2–96.3) and IgM seropositivity was 35.5% (95% CI = 19.22–54.6). From the partially vaccinated individuals, IgG seropositivity was 76.9% (95% CI = 46.2–95.0) with 53.8% (95% CI = 25.1–80.8) IgM seropositivity. Out of the participants that had not been vaccinated at all, IgG seroprevalence was 70.4% (95% CI 56.4–82.0) with 20.4% (95% CI 10.6–33.5) seropositivity of IgM antibodies.

## PCR positivity and vaccination status

When considering PCR positivity and COVID-19 vaccination status, one third of the individuals were positive (33.7%, 95 CI 24.3–44.1) while two thirds were negative (66.3%, 95% CI 55.9–

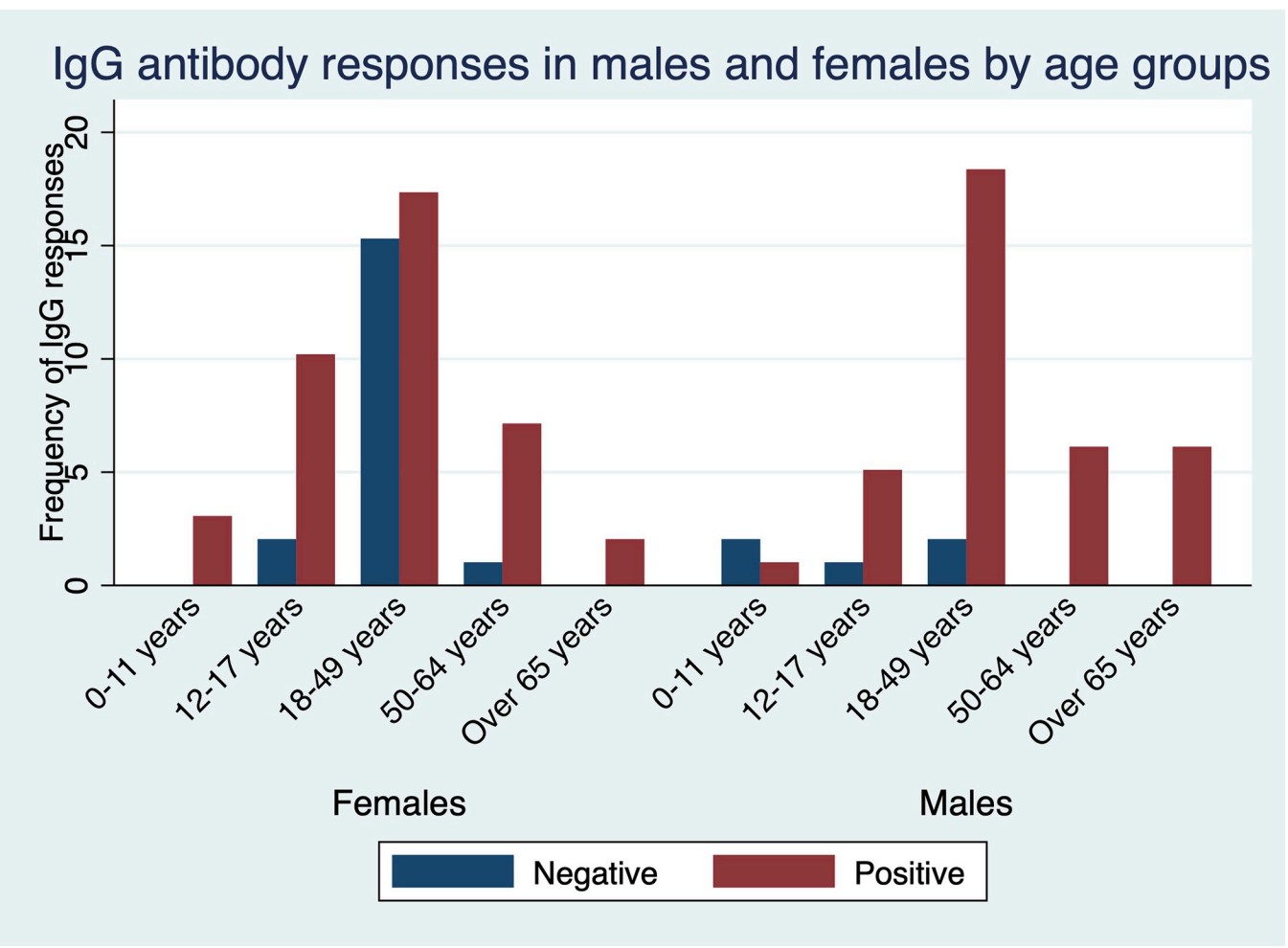

**Fig 1. Seroprevalence of infection-induced SARS-CoV-2 IgG antibodies, by gender and age group—Kisumu and Siaya counties, Kenya, December 2021–March 2022.**

75.7). The 32 positive individuals include 12(37.5%) fully vaccinated, 8(25%) partially vaccinated and 12(37.5%) unvaccinated. A binomial logistic regression was run to understand the effects of PCR positivity on the IgG and IgM seropositivity. SARs-CoV-2 PCR positivity did not significantly predict IgG (p = 0.469 [95% CI 0.514–4.230]) and IgM (p = 0.964 [95% CI 0.380–2.516]) positivity.

## Discussion

With the poor uptake of COVID-19 vaccines in African settings amidst the easing of restrictions on movements and other containment measures, there is need to understand the antibody responses in the population. The present study aimed to describe the anti-SARs-CoV-2 IgG and IgM antibody responses during the COVID-19 pandemic in the period between December 2021 and March 2022 in western Kenya. Generally, we found high levels of IgG and IgM antibody against SARS-CoV-2 in the population corroborating recent and previous findings from population-based surveys in Kenya [11, 15, 16]. As previously observed, this was despite the low vaccination rate with only about one third of the population receiving the full vaccination [17]. This implies that most of the population had been exposed to the COVID-19

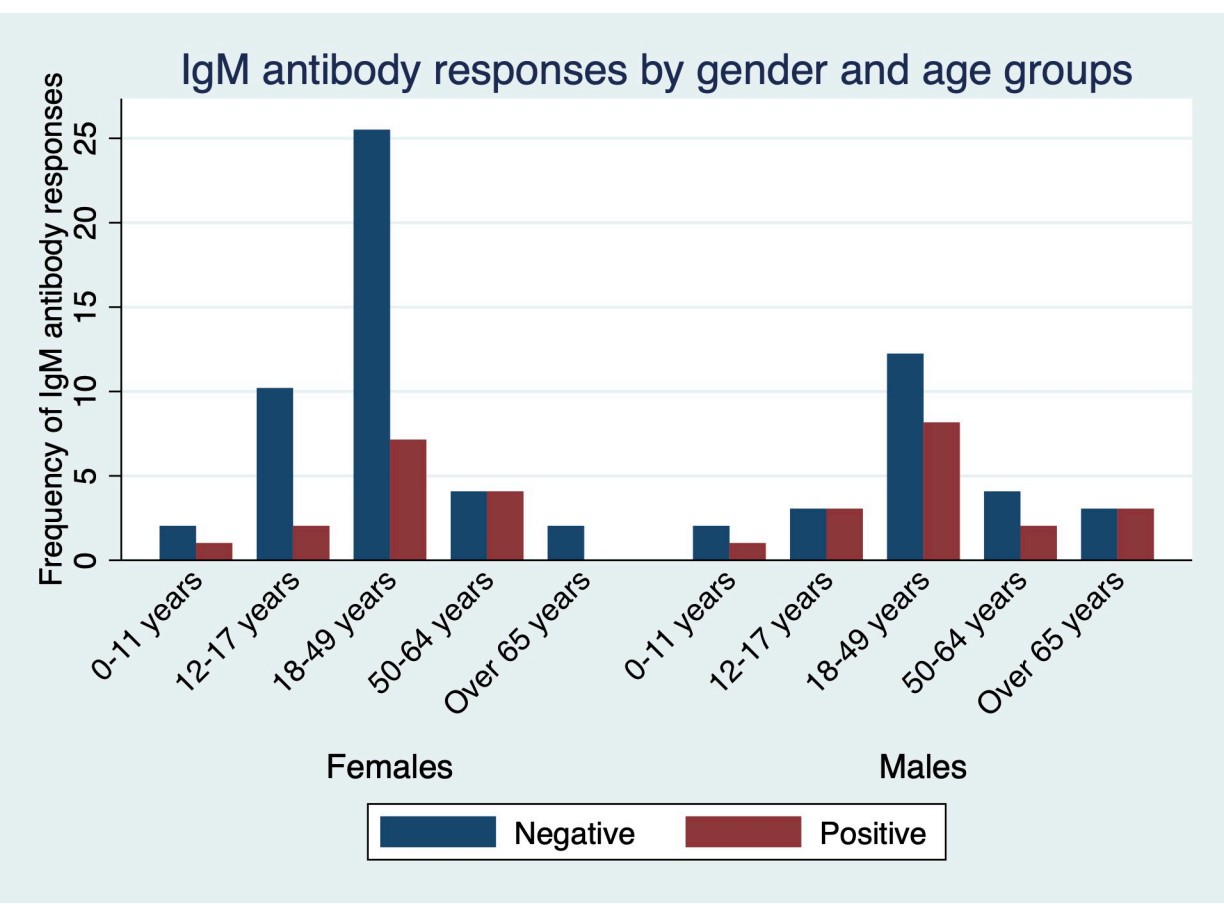

**Fig 2. Seroprevalence of infection-induced SARS-CoV-2 IgM antibodies, by gender and age group—Kisumu and Siaya counties, Kenya, December 2021–March 2022.**

virus during this period and had raised antibodies against the infection. This is suggestive that the population may be heading to herd immunity, but this should not lead to vaccine complacency. It is instructive to note, that the Kenyan government has prioritized vaccination of the entire population with first and booster doses readily available in public health facilities.

A large proportion, 26(81.2%) of the 32 PCR positives had SARS-COV-2 IgG antibodies suggestive of a robust immune response. However, both vaccine induced and natural immunity are imperfect with breakthrough infections reported across diverse COVID-19 transmission settings [18–21]. While COVID-19 vaccines elicit high levels of protection from symptomatic disease, this wanes over time necessitating booster doses to restore effectiveness [22]. Other studies have reported that immunogenicity may reduce despite of high IgG and neutralizing antibody levels [23]. Its noteworthy that, Kenya recorded its first case of the Omicron variant, besides the Delta variant during the study period. In addition, it's been suggested that humoral responses generated by vaccination may not be good enough to protect against Omicron infection [24] and that Omicron escapes the vast majority of existing SARS-COV-2 neutralizing antibodies [25].

When considering IgG antibody responses and age groups, seroprevalence peaked in adults aged 18–49 years consistent with other studies in Kenya that reported higher seroprevalence amongst adults [10]. It is plausible that the adults have an expanded immunological memory

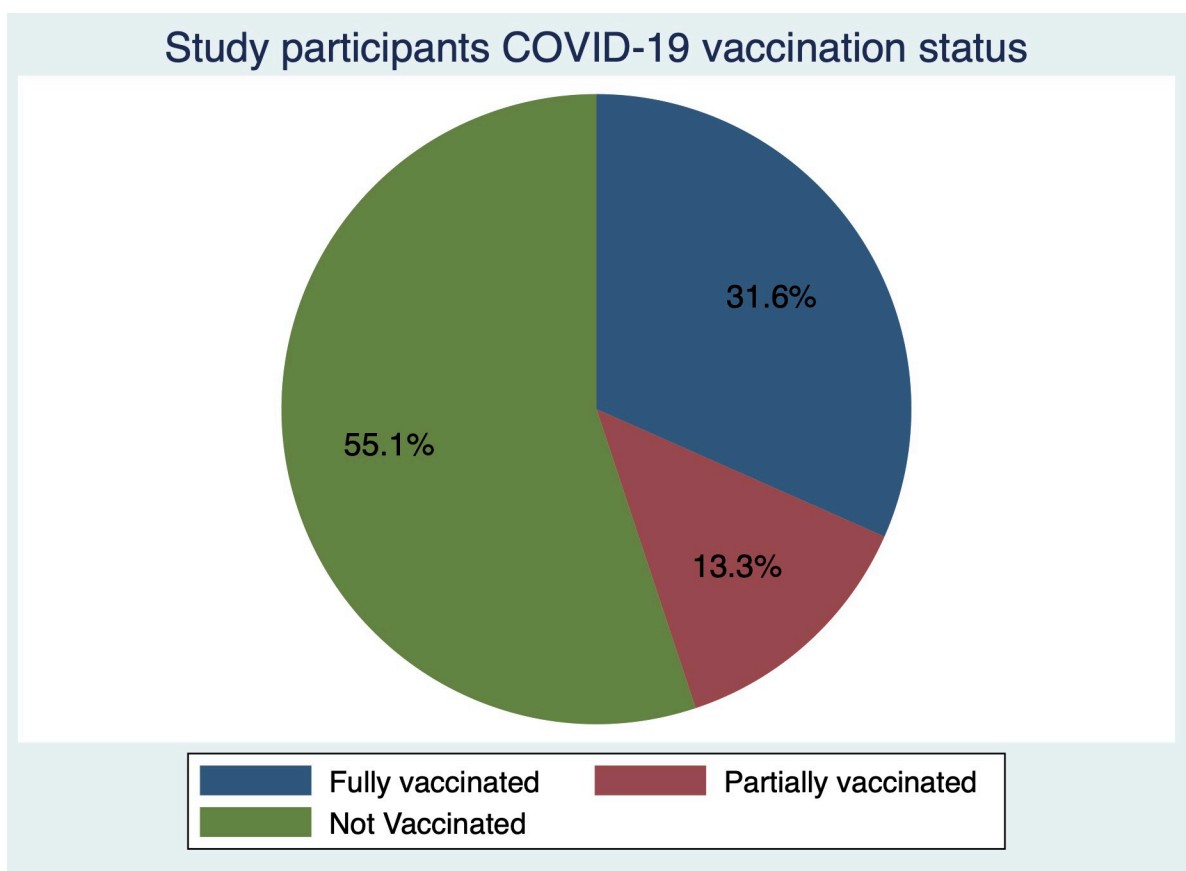

**Fig 3. Vaccination status of the study participants.**

driven by their catalog of memory B and T cells [26]. Adults older than 65 years of age had a lower antibody response against the SARS-CoV-2 compared to the other age groups in line with existing reports [9, 11, 12]. Increased comorbidities reported in the older adults may lead to the aging immune system not mounting a robust response.

Interestingly, we recorded a higher seroprevalence of IgM antibodies in Siaya County a more rural set up compared to Kisumu County a more urbanized town. This was in line with the higher SARS-CoV-2 RT-qPCR positivity of samples from Siaya county in contrast to Kisumu County during the sample collection period. This finding is consistent with other studies in Kenya that have reported marked geographic variation in seropositivity [10, 15]. Similar observations of variations in seroprevalence have also been documented across Africa [27, 28]. As the IgM antibody responses are short lived and useful in detecting recent infections [29], this suggest an ongoing community transmission in Siaya during the study period. The earlier and robust IgM responses days after onset of symptoms were linked to virus control. The similar profiles of IgG during the same period in the two counties corroborates its role as a more persistent antibody [30].

This study was limited by its recruitment of participants attending the hospitals, as it is possible that seroprevalence was overestimated due to selection bias. Additionally, the results may not be generalizable to the whole population as the samples may not have been representative. Consequently, representative longitudinal studies that follow individuals over a longer time span are needed to fully understand the SARS-CoV-2 antibody profiles and dynamics.

## Conclusions

Despite the low number of either fully or partially vaccinated individuals against SARS-CoV-2, the seroprevalence of IgG and IgM antibodies was high. The finding suggests that many study participants were already infected with the virus than what the PCR testing could cover.

## Supporting information

**S1 File. Raw demographics and laboratory data for Kisumu and Siaya counties, western, Kenya December 2021 to March 2022.**
(XLSX)

## Acknowledgments

We thank the study team for administrative and technical support. We appreciate the support received from the Kisumu and Siaya Counties health officers during study procedures. We are grateful to the study participants who took part in the study.

## Author Contributions

**Conceptualization:** Shehu Shagari Awandu, Alfred Ochieng Ochieng, Benson Onyango, Richard Odongo Magwanga, Fredrick Ogutu, Benson Estambale.

**Data curation:** Shehu Shagari Awandu.

**Formal analysis:** Shehu Shagari Awandu.

**Funding acquisition:** Shehu Shagari Awandu, Richard Odongo Magwanga, Benson Estambale.

**Investigation:** Shehu Shagari Awandu, Jim Seth Katieno, Joy Nyangasi Kirungu, Francis Orata, Justine Achieng, Bonface Oure, Regina Nyunja, Eric M. O. Muok, Stephen Munga, Benson Estambale.

**Methodology:** Shehu Shagari Awandu, Pamela Were, Angeline Atieno Ochung', Fredrick Okumu, Marceline Adhiambo Oloo, Jim Seth Katieno, Shirley Lidechi, Fredrick Ogutu, Dorothy Awuor, Eric M. O. Muok, Stephen Munga, Benson Estambale.

**Visualization:** Shehu Shagari Awandu.

**Writing – original draft:** Shehu Shagari Awandu, Fredrick Ogutu.

**Writing – review & editing:** Shehu Shagari Awandu, Alfred Ochieng Ochieng, Benson Onyango, Richard Odongo Magwanga, Pamela Were, Angeline Atieno Ochung', Fredrick Okumu, Marceline Adhiambo Oloo, Jim Seth Katieno, Shirley Lidechi, Dorothy Awuor, Francis Orata, Justine Achieng, Bonface Oure, Regina Nyunja, Eric M. O. Muok, Stephen Munga, Benson Estambale.

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
