## [Decision Letter · Decision Letter 0]

20 Sep 2022

PONE-D-22-20904High seroprevalence of immunoglobulin G (IgG) and IgM antibodies to SARS-CoV-2 in asymptomatic and symptomatic individuals amidst vaccination roll-out in western KenyaPLOS ONE

Dear Dr. Awandu,

Thank you for submitting your manuscript to PLOS ONE. After careful consideration, we feel that it has merit but does not fully meet PLOS ONE’s publication criteria as it currently stands. Therefore, we invite you to submit a revised version of the manuscript that addresses the points raised during the review process. The Authors should address to the criticisms detected by the REferees in order to clarify some inconsistencies. Furhtermore, they should check once again the number indicated in the manuscript. 

We look forward to receiving your revised manuscript.

Kind regards,

Adriana Calderaro

Academic Editor

PLOS ONE

Journal Requirements:

Reviewers' comments:

Reviewer's Responses to Questions

**Comments to the Author**

1. Is the manuscript technically sound, and do the data support the conclusions?

Reviewer #1: Yes

Reviewer #2: No

2. Has the statistical analysis been performed appropriately and rigorously? 

Reviewer #1: Yes

Reviewer #2: Yes

3. Have the authors made all data underlying the findings in their manuscript fully available?

Reviewer #1: Yes

Reviewer #2: Yes

4. Is the manuscript presented in an intelligible fashion and written in standard English?

Reviewer #1: Yes

Reviewer #2: Yes

5. Review Comments to the Author

Reviewer #1: I think that this manuscript should be accepted with minor revision in introduction or part of the Demographics in the Results section. I suggest that authors add data on numbers of registered cases from the beginning of the pandemic in these two counties and also data on number of registered cases for the observed period in both counties. Possibly a figure could be drawn on incidence of registered cases and specific incidences by observed age groups as well. This is a minor effort and i belive it would fir well with the rest of the text and give it more clear overall picture

Reviewer #2: In this manuscript, Awandu et al. report the seroprevalence estimate in 98 individuals presenting to Kisumu and Siaya Counties referral hospitals in western Kenya between December 2021-March 2022. The authors measured SARS-CoV2 IgG and IgM antibodies in blood specimens and detected viral RNA in nasopharyngeal samples using commercial kits following manufacturer instructions. Of the 98 subjects, the authors report that 44 were vaccinated fully (n=31) or partially (n=13). They say high IgG seropositivity in both vaccinated (86.7%) and unvaccinated (70.3%) individuals and claim that PCR testing did not predict the seropositivity within the study period. The study has several weaknesses, some of which are listed below.

In the serology data sheet, the authors report a total of 32 PCR positives. This includes 12 (38.7%) PCR positives among 31 fully vaccinated, 8 (61.5%) PCR positives among 13 partially vaccinated, and 11 (20.3%) PCR positives among 54 unvaccinated individuals. Interestingly, 26 (83.8%) of the 31 PCR positives have been reported to have SARS-CoV-2 IgG antibodies. While breakthrough infections are possible, it is unclear why such a large proportion of PCR positives have SARS-CoV-2 IgG Abs. It is also unclear why PCR positives are higher among vaccinated than unvaccinated (47.7% (23 of 44) Vs. 20.3% (11 of 54).

The authors state that they have used commercial kits following manufacturer instructions. It is unclear if the authors have tested pre-pandemic samples to validate the commercial assay specificity.

Authors report the IgG and IgM results as ratios. However, the methods described in line 124 are unclear about how ratios were calculated from optical densities.

The number of vaccinated individuals reported in the abstract (n=30) and the serology data sheet (n=31) do not match.

Authors report that about 26 of 98 were asymptomatic. It will be helpful to clarify how asymptomatic were referred to the hospitals. Are these asymptomatic people exposed to people previously infected with SARS-CoV-2?

6. PLOS authors have the option to publish the peer review history of their article (what does this mean?). If published, this will include your full peer review and any attached files.

Reviewer #1: No

Reviewer #2: No

---

## [Author Response · Author response to Decision Letter 0]

24 Oct 2022

Response to Reviewers

Dear Editors and Reviewers,

Thank you for giving me the opportunity to submit a revised draft of our manuscript titled High seroprevalence of Immunoglobulin G (IgG) and IgM antibodies to SARS-CoV-2 in asymptomatic and symptomatic individuals amidst vaccination roll-out in western Kenya to Plos One Journal. We appreciate you and the reviewers for dedicating your precious time in reviewing our manuscript and providing valuable feedback. We have been able to incorporate changes to reflect most of the suggestions provided by the reviewers. We hope that we have satisfactorily addressed them and that the manuscript is now suitable for publication.

Sincerely,

On behalf of all authors,

Shehu Shagari Awandu

Here is a point-by-point response to the reviewers’ comments and concerns.

Comments from the Academic Editor

Comment 1. Please ensure that your manuscript meets PLOS ONE's style requirements, including those for file naming. 

Response. Thank you very much for these comments. We have ensured that the manuscript meets PLOS ONE's style requirements, including those for file naming. We hopefully have no divergences from the style requirements now. 

Comment 2. We note that the grant information you provided in the ‘Funding Information’ and ‘Financial Disclosure’ sections do not match. 

Thanks for pointing this out. We have provided the correct grant numbers. The study was supported by the National Research Foundation, South Africa under the COVID-19 Africa Rapid Grant Fund Grant Number COV19200616532700 awarded to Benson Estambale. “

Comment 3. Please include your full ethics statement in the ‘Methods’ section of your manuscript file. In your statement, please include the full name of the IRB or ethics committee who approved or waived your study, as well as whether or not you obtained informed written or verbal consent. If consent was waived for your study, please include this information in your statement as well.

Response: Thanks for the comment. We have included the full ethics statement on and it now reads as follows

“Ethical consideration

Ethical approval for the study was granted by Jaramogi Oginga Odinga University of Science and Technology (ERC/21/5/21-4), and a research license granted from Kenya National Commission of Science and Technolog, (NACOSTI/P/22/17545). Administrative approval was provided by the county governments of Kisumu and Siaya. All participants provided written informed consent or assent before enrollment.”

Comment 4. We note that Figure 1 in your submission contain [map/satellite] images which may be copyrighted. All PLOS content is published under the Creative Commons Attribution License (CC BY 4.0), which means that the manuscript, images, and Supporting Information files will be freely available online, and any third party is permitted to access, download, copy, distribute, and use these materials in any way, even commercially, with proper attribution. For these reasons, we cannot publish previously copyrighted maps or satellite images created using proprietary data, such as Google software (Google Maps, Street View, and Earth).

Response: Thanks for the comment. We have removed the figure

Comments from Reviewer #1

Comments 1: I think that this manuscript should be accepted with minor revision in introduction or part of the Demographics in the Results section. 

Response: Thank you very much 

Comment 2: I suggest that authors add data on numbers of registered cases from the beginning of the pandemic in these two counties and also data on number of registered cases for the observed period in both counties. Possibly a figure could be drawn on incidence of registered cases and specific incidences by observed age groups as well. This is a minor effort and i belive it would fir well with the rest of the text and give it more clear overall picture.

Response: Thank you very much for this pointing this out. We have updated the manuscript and provided data on the number of registered cases from the beginning of the pandemic, and during our study period. We have included the Kenya Ministry of Health reference that provides a comprehensive COVID-19 data for the whole country. 

Response to Reviewer 2

In this manuscript, Awandu et al. report the seroprevalence estimate in 98 individuals presenting to Kisumu and Siaya Counties referral hospitals in western Kenya between December 2021-March 2022. The authors measured SARS-CoV2 IgG and IgM antibodies in blood specimens and detected viral RNA in nasopharyngeal samples using commercial kits following manufacturer instructions. Of the 98 subjects, the authors report that 44 were vaccinated fully (n=31) or partially (n=13). They say high IgG seropositivity in both vaccinated (86.7%) and unvaccinated (70.3%) individuals and claim that PCR testing did not predict the seropositivity within the study period. The study has several weaknesses, some of which are listed below.

Reviewer: In the serology data sheet, the authors report a total of 32 PCR positives. This includes 12 (38.7%) PCR positives among 31 fully vaccinated, 8 (61.5%) PCR positives among 13 partially vaccinated, and 11 (20.3%) PCR positives among 54 unvaccinated individuals. Interestingly, 26 (83.8%) of the 31 PCR positives have been reported to have SARS-CoV-2 IgG antibodies. While breakthrough infections are possible, it is unclear why such a large proportion of PCR positives have SARS-CoV-2 IgG Abs. It is also unclear why PCR positives are higher among vaccinated than unvaccinated (47.7% (23 of 44) Vs. 20.3% (11 of 54).

Response: Thanks for your comments, that have helped us improve the section. We have revised the discussion to include a paragraph providing a possible explanation for the high rate of IgG and PCR positivity. 

“A large proportion, 26(81.2%) of the 32 PCR positives had SARS-COV-2 IgG antibodies suggestive of a robust immune response. However, both vaccine induced, and natural immunity are imperfect with breakthrough infections reported across diverse COVID-19 transmission settings (18-21). Other studies have reported that immunogenicity may reduce despite of high IgG and neutralizing antibody levels (22). Its noteworthy that, Kenya recorded its first case of the omicron variant during the study period, and this may explain our findings. In addition, it’s been suggested that humoral responses generated by vaccination may not be good enough to protect against omicron infection (23) and that omicron escapes the vast majority of existing SARS-COV-2 neutralizing antibodies (24).”

Reviewer comments: The authors state that they have used commercial kits following manufacturer instructions. It is unclear if the authors have tested pre-pandemic samples to validate the commercial assay specificity.

Response: Thank you very much for pointing this out. This has been expounded in the manuscript. 

Reviewer: Authors report the IgG and IgM results as ratios. However, the methods described in line 124 are unclear about how ratios were calculated from optical densities.

Response: We agree with the reviewer that the methods were unclear. We have included a detailed explanation of how the ratios are computed in the manuscript.

Reviewer: The number of vaccinated individuals reported in the abstract (n=30) and the serology data sheet (n=31) do not match.

Response: Thank you for pointing this out. We admit our error. We have corrected the error(n=31) and now both the abstract and serology datasheet match. We have similarly gone through the entire manuscript and updated the figures based on the data sheet

Reviewer: Authors report that about 26 of 98 were asymptomatic. It will be helpful to clarify how asymptomatic were referred to the hospitals. Are these asymptomatic people exposed to people previously infected with SARS-CoV-2?

Response: Thanks for pointing this out, we have clarified this on the manuscripts and it now reads as follows.

“About 26.5% (26/98) of participants were asymptomatic, however, they were referred to the hospital for surveillance testing as some were contacts of people infected with SARS-CoV-2.”

---

## [Decision Letter · Decision Letter 1]

24 Nov 2022

High seroprevalence of immunoglobulin G (IgG) and IgM antibodies to SARS-CoV-2 in asymptomatic and symptomatic individuals amidst vaccination roll-out in western Kenya

PONE-D-22-20904R1

Dear Dr. Awandu,

We’re pleased to inform you that your manuscript has been judged scientifically suitable for publication and will be formally accepted for publication once it meets all outstanding technical requirements.

Kind regards,

Adriana Calderaro

Academic Editor

PLOS ONE

Additional Editor Comments (optional):

Reviewers' comments:

Reviewer's Responses to Questions

**Comments to the Author**

1. If the authors have adequately addressed your comments raised in a previous round of review and you feel that this manuscript is now acceptable for publication, you may indicate that here to bypass the “Comments to the Author” section, enter your conflict of interest statement in the “Confidential to Editor” section, and submit your "Accept" recommendation.

Reviewer #1: All comments have been addressed

Reviewer #2: All comments have been addressed

2. Is the manuscript technically sound, and do the data support the conclusions?

Reviewer #1: Yes

Reviewer #2: Yes

3. Has the statistical analysis been performed appropriately and rigorously? 

Reviewer #1: Yes

Reviewer #2: Yes

4. Have the authors made all data underlying the findings in their manuscript fully available?

Reviewer #1: Yes

Reviewer #2: Yes

5. Is the manuscript presented in an intelligible fashion and written in standard English?

Reviewer #1: Yes

Reviewer #2: Yes

6. Review Comments to the Author

Reviewer #1: I think all issues have been addressed and paper should be accepted. Paper is well written, all reviewers questions were met. Results are described clearly, Discussion is well referenced and limitations support the conclusion.

Reviewer #2: (No Response)

7. PLOS authors have the option to publish the peer review history of their article (what does this mean?). If published, this will include your full peer review and any attached files.

Reviewer #1: No

Reviewer #2: No

---

## [Editor Report · Acceptance letter]

13 Dec 2022

PONE-D-22-20904R1 

High seroprevalence of Immunoglobulin G (IgG) and IgM antibodies to SARS-CoV-2 in asymptomatic and symptomatic individuals amidst vaccination roll-out in western Kenya 

Dear Dr. Awandu:

I'm pleased to inform you that your manuscript has been deemed suitable for publication in PLOS ONE. Congratulations! Your manuscript is now with our production department. 

Kind regards, 

on behalf of

MD, PhD, Associate Professor Adriana Calderaro 

Academic Editor

PLOS ONE